# Association between Geriatric Nutrition Risk Index and The Presence of Sarcopenia in People with Type 2 Diabetes Mellitus: A Cross-Sectional Study

**DOI:** 10.3390/nu13113729

**Published:** 2021-10-22

**Authors:** Fuyuko Takahashi, Yoshitaka Hashimoto, Ayumi Kaji, Ryosuke Sakai, Yuka Kawate, Takuro Okamura, Noriyuki Kitagawa, Hiroshi Okada, Naoko Nakanishi, Saori Majima, Takafumi Senmaru, Emi Ushigome, Mai Asano, Masahide Hamaguchi, Masahiro Yamazaki, Michiaki Fukui

**Affiliations:** 1Department of Endocrinology and Metabolism, Graduate School of Medical Science, Kyoto Prefectural University of Medicine, Kyoto 602-8566, Japan; fuyuko-t@koto.kpu-m.ac.jp (F.T.); kaji-a@koto.kpu-m.ac.jp (A.K.); sakaryo@koto.kpu-m.ac.jp (R.S.); yukawate@koto.kpu-m.ac.jp (Y.K.); d04sm012@koto.kpu-m.ac.jp (T.O.); nori-kgw@koto.kpu-m.ac.jp (N.K.); conti@koto.kpu-m.ac.jp (H.O.); naoko-n@koto.kpu-m.ac.jp (N.N.); saori-m@koto.kpu-m.ac.jp (S.M.); semmarut@koto.kpu-m.ac.jp (T.S.); emis@koto.kpu-m.ac.jp (E.U.); maias@koto.kpu-m.ac.jp (M.A.); mhama@koto.kpu-m.ac.jp (M.H.); masahiro@koto.kpu-m.ac.jp (M.Y.); michiaki@koto.kpu-m.ac.jp (M.F.); 2Department of Diabetology, Kameoka Municipal Hospital, 1-1 Noda, Shinochoshino, Kyoto 621-8585, Japan; 3Department of Diabetes and Endocrinology, Matsushita Memorial Hospital, 5-55 Sotojimacho, Osaka 570-8540, Japan

**Keywords:** geriatric nutritional risk index, sarcopenia, type 2 diabetes mellitus

## Abstract

The aim of this cross-sectional study was to examine the association between the geriatric nutritional risk index (GNRI) and the prevalence of sarcopenia in people with type 2 diabetes (T2DM). Having both low handgrip strength (<28 kg for men and <18 kg for women) and low skeletal muscle mass index (<7.0 kg/m^2^ for men and <5.7 kg/m^2^ for women) was diagnosed as sarcopenia. GNRI was estimated by the formula as below: GNRI = (1.489 × serum albumin level [g/L]) + (41.7 × [current body weight (kg)/ideal body weight (kg)]). Participants were dichotomized on the basis of their GNRI scores (GNRI < 98, low; or GNRI ≥ 98, high). Among 526 people (301 men and 225 women) with T2DM, the proportions of participants with sarcopenia and low GNRI were 12.7% (*n* = 67/526) and 5.1% (*n* = 27/526), respectively. The proportion of sarcopenia in participants with low-GNRI was higher than that with high GNRI (44.4% [*n* = 12/27] vs. 11.0% [*n* = 55/499], *p* < 0.001). The GNRI showed positive correlations with handgrip strength (*r* = 0.232, *p* < 0.001) and skeletal muscle mass index (*r* = 0.514, *p* < 0.001). Moreover, low GNRI was related to the prevalence of sarcopenia (adjusted odds ratio, 4.88 [95% confidence interval: 1.88–12.7], *p* = 0.001). The GNRI, as a continuous variable, was also related to the prevalence of sarcopenia (adjusted odds ratio, 0.89 [95% confidence interval: 0.86–0.93], *p* < 0.001). The present study revealed that low GNRI was related to the prevalence of sarcopenia.

## 1. Introduction

The population of people with type 2 diabetes mellitus (T2DM) is globally on the rise [1]. Sarcopenia is a potentially greater public health concern [2], with the aging population. The Asian Working Group for Sarcopenia states that the characteristics of sarcopenia are low muscle strength, muscle mass loss, and low physical performance [2]. The mechanisms underlying the muscle mass loss differ between healthy individuals and people with diabetes. Muscle catabolism is enhanced in individuals with T2DM due to insulin resistance and attenuated insulin signaling [3]. In fact, T2DM has been related to a 1.55-fold higher risk of sarcopenia in older adults than in the general population [4]. Therefore, sarcopenia in people with T2DM requires more attention than those without diabetes.

There is a close association between malnutrition and sarcopenia [5]. The geriatric nutritional risk index (GNRI) is a nutritional status indicator, and a simple and accurate screening instrument that comprises a few objective factors, such as weight, height, and serum albumin [6]. Previous studies revealed that the GNRI was related to all-cause mortality [7] and that the GNRI was associated with sarcopenia in older people on hospitalization [8] and patients undergoing hemodialysis [9]. A negative correlation was found between serum albumin levels and increased extracellular fluid level [10]. Body weight is also influenced by the state of hydration, but the variations in the state of hydration strongly contrast with the variations in serum albumin levels. Using both indicators in the GNRI, the confounding variables such as the state of hydration can be minimized. Similarly, serum albumin levels are associated with comorbidities, which are related to malnutrition. Therefore, the GNRI, which considers both serum albumin levels and body weight, is suitable for assessing nutritional status. However, the association between the GNRI and the presence of sarcopenia in people with T2DM remains unclear. Thus, this cross-sectional study of people with T2DM researched the association between the GNRI and sarcopenia.

## 2. Method

### 2.1. Study Participants

The data from the ongoing KAMOGAWA-DM cohort study, started in 2014, were used in this cross-sectional study [11]. This cohort study included outpatients of the Department of Endocrinology and Metabolism, Kyoto Prefectural University of Medicine Hospital (Kyoto, Japan), and the Department of Diabetology, Kameoka Municipal Hospital (Kameoka, Japan). In this study, people with T2DM with body composition measurements from January 2015 to August 2021 were included. People with no data on handgrip strength and serum albumin were excluded. Approval was obtained from the local research ethics committee (no. RBMR-E-466-6), and this study was performed in accordance with the Declaration of Helsinki. Informed consent was received in writing from all the participants.

### 2.2. Data Collection

A standardized questionnaire was used to assess the duration of diabetes, the family history of diabetes, the status of smoking, and the habit of exercise. Participants were categorized into two groups: non-smokers and smokers. Regular exercise habits were defined as regularly playing any kind of sport >1×/week.

Overnight fasting venous blood samples were gathered from participants, and the levels of high-density lipoprotein cholesterol, triglycerides, fasting plasma glucose, C-reactive protein (CRP), uric acid, creatinine (Cr), and albumin were assessed. Hemoglobin A1c (HbA1c) levels were evaluated using high-performance liquid chromatography and presented in units of the National Glycohemoglobin Standardization Program. The estimated glomerular filtration rate (eGFR) was estimated based on the formula of the Japanese Society of Nephrology: eGFR = 194 × Cr^−1.094^ × age^−0.287^ (mL/min/1.73 m^2^) (×0.739, if woman) [12]. Blood pressure was measured automatically in a quiet room after a 5 min rest.

Medication data, including antidiabetics drugs (insulin, sodium glucose cotransporter-2 (SGLT2) inhibitors and glucagon-like peptide-1 (GLP-1) agonists), and antihypertensive drugs were sourced from the medical records. Hypertension was defined as usage of antihypertensive drugs, systolic blood pressure of ≥140 mmHg, and/or diastolic blood pressure of ≥90 mmHg [13].

Dietary intake was evaluated by brief-type self-administered diet history questionnaire (BDHQ), which estimated a dietary intake of 58 items over the past month [14]. Total energy (kcal/day); total protein (g/day); fat (g/day); and carbohydrate (g/day) intakes were obtained by the BDHQ. Total energy (kcal/ideal body weight/day), total protein (g/ideal body weight/day), fat (g/ideal body weight/day), and carbohydrate (g/ideal body weight/day) intakes were calculated.

### 2.3. Definition of Sarcopenia

Body composition was assessed with a multifrequency impedance body composition analyzer (InBody 720, InBody Japan, Tokyo, Japan), which correlates well with dual-energy X-ray absorptiometry [15]. Using the data of this analyzer, body weight (BW, kg), body fat mass (kg), and appendicular muscle mass (kg) were gathered. Then, body mass index (BMI, kg/m^2^) or skeletal muscle mass index (SMI, kg/m^2^) was estimated by BW (kg) or appendicular muscle mass (kg) divided by height squared (m^2^) [15].

The maximum value of handgrip strength of both hands, which was measured by handgrip dynamometer (Smedley, Takei Scientific Instruments Co., Ltd., Niigata, Japan), was employed in the analysis.

Sarcopenia was defined by SMI and handgrip strength [2]. Having both low muscle strength (handgrip strength: <28 kg for men and <18 kg for women) and low skeletal muscle mass (SMI: <7.0 kg/m^2^ for men and <5.7 kg/m^2^ for women) was classified as sarcopenia [2].

### 2.4. Definition of Geriatric Nutritional Risk Index

The GNRI was calculated using the formula as below: GNRI = (1.489 × serum albumin level [g/L]) + (41.7 × [current body weight (kg)/ideal body weight (kg)]) [6]. In this study, ideal body weight was determined from the participant’s height and a BMI of 22 kg/m^2^ [16]. In accordance with previous studies [6,17], the participants were separated into two groups basing on their GNRI scores (GNRI low [<98] or GNRI high [≥98]).

### 2.5. Statistical Analyses

Data were expressed as means (standard deviation [SD]) or frequencies of potential confounding variables.

Participants were grouped into two groups based on their GNRI scores. Differences in continuous and categorical variables were evaluated using the Student’s *t*-test and chi-squared test, respectively. Correlations were analyzed using the Pearson’s correlation coefficient. Logistic regression analyses were conducted to obtain the odds ratio (OR) and 95% confidence interval (CI) for the GNRI on the prevalence of sarcopenia, after adjusting for age, sex, duration of diabetes, habit of smoking, habit of exercise, HbA1c, insulin treatment, usage of SGLT2 inhibitor, and usage of GLP-1 antagonist.

Statistical analyses were conducted using EZR (Saitama Medical Center, Jichi Medical University, Saitama, Japan) [18], a graphical user interface for R (The R Foundation for Statistical Computing, Vienna, Austria). *p* values of < 0.05 were considered significant.

## 3. Results

In total, 560 individuals with T2DM were assessed for body composition and enrolled in this study. We excluded 34 participants, of whom 25 were not assessed for handgrip strength, while 9 were not assessed for serum albumin levels; therefore, the final study population comprised 526 participants (301 men and 225 women; Figure 1). In addition, among 526 participants, 452 participants (433 participants with high GNRI and 19 participants with low GNRI) were surveyed about their dietary intake.

The clinical characteristics of the study participants are summarized in Table 1. The mean age or BMI was 67.1 ± 10.9 years or 24.4 ± 4.3 kg/m^2^ in all participants. The mean BMI was 23.9 ± 3.6 kg/m^2^ in men and 25.0 ± 5.0 kg/m^2^ in women. The proportions of participants with sarcopenia and low GNRI were 12.7% (*n* = 67/526) and 5.1% (*n* = 27/526), respectively. The proportions of participants with sarcopenia and low GNRI were 14.0% (*n* = 42/301) and 6.3% (*n* = 19/301) in men, and 11.1% (*n* = 25/225) and 3.6% (*n* = 8/225) in women, respectively.

Table 2 shows the participants’ clinical characteristics according to their GNRI scores. Participants with low GNRI were older than those with high GNRI (73.2 ± 7.9 vs. 66.7 ± 10.9 years, *p* = 0.002). The BMI in participants with low GNRI was lower than that in participants with high GNRI (19.3 ± 2.5 vs. 24.7 ± 4.2 kg/m^2^, *p* < 0.001). The levels of HbA1c (66.5 ± 25.0 vs. 56.6 ± 12.2 mmol/mol, *p* < 0.001) and CRP (16,025.9 ± 30,822.4 vs. 2115.7 ± 7863.2 ug/L, *p* < 0.001) in participants with low GNRI were higher than those in participants with high GNRI. The proportion of patients exclusively on multi-injection treatment with insulin in low GNRI was higher than that in high GNRI (22.2% [*n* = 6/27] vs. 4.2% [*n* = 21/499], *p* < 0.001). The prevalence of low skeletal muscle mass (66.7% [*n* = 18/27] vs. 22.0% [*n* = 110/499], *p* < 0.001), low muscle strength (59.3% [*n* = 16/27] vs. 23.0% [*n* = 115/499], *p* < 0.001), and sarcopenia (44.4% [*n* = 12/27] vs. 11.0% [*n* = 55/499], *p* < 0.001) in participants with low GNRI were higher than those in participants with high GNRI. There was no difference of habitual diet intakes between participants with high GHRI and low GNRI.

Appendix A presents the characteristics differences between the low GNRI group and high GNRI group of men and women. Men with low GNRI were significantly older than those with high GNRI (73.5 ± 8.4 vs. 67.3 ± 10.9 years, *p* = 0.016), and women with low GNRI tended to be older than those with high GNRI, although the differences were not significant (72.5 ± 7.0 vs. 65.9 ± 10.9 years, *p* = 0.092). The HbA1c levels in both men (63.9 ± 21.6 vs. 56.6 ± 11.9 mmol/mol, *p* = 0.016) and women (72.5 ± 32.5 vs. 56.5 ± 12.6 mmol/mol, *p* = 0.001) with low GNRI were higher than those with high GNRI. The CRP levels in both men (15,389.5 ± 31,230.6 vs. 1616.0 ± 3694.2 ug/L, *p* < 0.001) and women (17,537.5 ± 31,528.0 vs. 2747.6 ± 11,063.3 ug/L, *p* < 0.001) with low GNRI were higher than those with high GNRI. Moreover, the prevalence of sarcopenia in both men (42.1% [*n* = 8/19] vs. 12.1% [*n* = 34/282], *p* < 0.001) and women (50.0% [*n* = 4/8] vs. 9.8% [*n* = 21/217], *p* = 0.003) with low GNRI was higher than those in men and women with high GNRI.

The correlation between the GNRI and SMI or handgrip strength is shown in Figure 2. GNRI was correlated with SMI (*r* = 0.514, *p* < 0.001) and handgrip strength (*r* = 0.232, *p* < 0.001) in all participants. In both men and women, GNRI was positively correlated with SMI (men: *r* = 0.627, *p* < 0.001; women: *r* = 0.726, *p* < 0.001) and handgrip strength (men: *r* = 0.447, *p* < 0.001; women: *r* = 0.325, *p* < 0.001).

The association between the GNRI and the presence of sarcopenia is presented in Table 3. Low GNRI was related to the prevalence of sarcopenia (adjusted OR, 4.88 [95% CI: 1.88–12.7], *p* = 0.001).

The GNRI, as a continuous variable, was also related to the prevalence of sarcopenia (adjusted OR, 0.89 [95% CI: 0.86–0.93], *p* < 0.001) (Table 4). The relationship between the GNRI and the presence of sarcopenia according to sex is shown in Appendix A. Low GNRI was related to the presence of sarcopenia in both men (adjusted OR, 4.22 [95% CI: 1.26–14.1], *p* = 0.019) and women (adjusted OR, 11.3 [95% CI: 1.85–69.0], *p* = 0.009). Moreover, the GNRI, as a continuous variable, was associated with the presence of sarcopenia in both men (adjusted OR, 0.90 [95% CI: 0.86–0.95] *p* < 0.001) and women (adjusted OR, 0.86 [95% CI: 0.80–0.93], *p* < 0.001).

## 4. Discussion

We initially examined the relationship between the GNRI and the prevalence of sarcopenia in people with T2DM. The primary finding of this study was that the GNRI was associated with the presence of sarcopenia in people with T2DM after adjusting for covariates.

Nutritional status and sarcopenia are related to each other in their pathophysiology [19]. Previous studies have shown the association between the GNRI and the presence of sarcopenia in people undergoing hemodialysis [9] and in hospitalized older adults [8]. In addition, the GNRI was regarded as a tool to predict muscle dysfunction in general older population [20] and strongly associated with muscle function, expressed as handgrip strength, in a large sample of institutionalized older people [21,22].

The possible examinations of the relationship between the GNRI and sarcopenia are described below. Sarcopenia occurs due to a decrease in protein synthesis and an increase in protein degradation caused by increased oxidative stress, decreased antioxidant defenses, and inflammation. Sarcopenia has been related to nuclear factor κB and protein kinase B signaling through secretion of tumor necrosis factor-α, transforming growth factor-β and interleukin-6 [23]. On the contrary, an association between the GNRI and inflammation has been reported [24,25], and the GNRI was negatively correlated with CRP [26], which corresponds to the results of the study. Chronic inflammation that occurs in T2DM induces muscle atrophy [27,28]. Therefore, the risk of sarcopenia in people with T2DM is higher than that in the general population [4]. Long duration of diabetes may be associated with a long period of chronic inflammation, and increased risk of sarcopenia. In this study, duration of diabetes in people with sarcopenia was longer than that in people without sarcopenia (21.0 ± 11.5 vs. 13.6 ± 9.7 years, *p* < 0.001). Furthermore, duration of diabetes was negatively correlated with SMI (*r* = −0.157, *p* < 0.001) and handgrip strength (*r* = −0.195, *p* < 0.001). In addition, there was a negative relationship between the GNRI and duration of diabetes (*r* = −0.291, *p* < 0.001) in this study. It is possible that people with long duration of diabetes have a long period of chronic inflammation and low GNRI, leading to sarcopenia. Moreover, malnutrition is a major risk factor for sarcopenia and plays as a driver of loss of muscle mass and function, which are the main features [5,29]. In fact, the GNRI, which is attributed to albumin and body weight, has been associated with acute and chronic malnutrition [7,30]. Therefore, there is an association between the GNRI and sarcopenia due to the presence of inflammation and malnutrition. Habitual dietary intakes were not associated with the GNRI, although it has been reported that nutritional status and habitual dietary intake were closely related [31]. This might be because that this was a cross-sectional study and the participants already received nutritional guidance.

There are several limitations in the present study. First, this study is an observational study. Thus, it is possible that unknown confounding factors are present. Second, the all participants’ dietary intake status was not assessed. Third, physical performance was not evaluated. Fourth, we did not assess the concentration of C-peptide in this study. Therefore, this study did not consider the capacity for insulin secretion. Finally, the present study was cross-sectional in nature. Therefore, the causative association between the GNRI and the prevalence of sarcopenia remains unclear.

## 5. Conclusions

This study revealed that the GNRI is related to the presence of sarcopenia in people with T2DM. Since nutritional status and sarcopenia are interrelated, it is essential to sustain a good nutritional status to prevent sarcopenia and to maintain muscle mass and muscle strength to prevent malnutrition.

## Figures and Tables

**Figure 1 nutrients-13-03729-f001:**
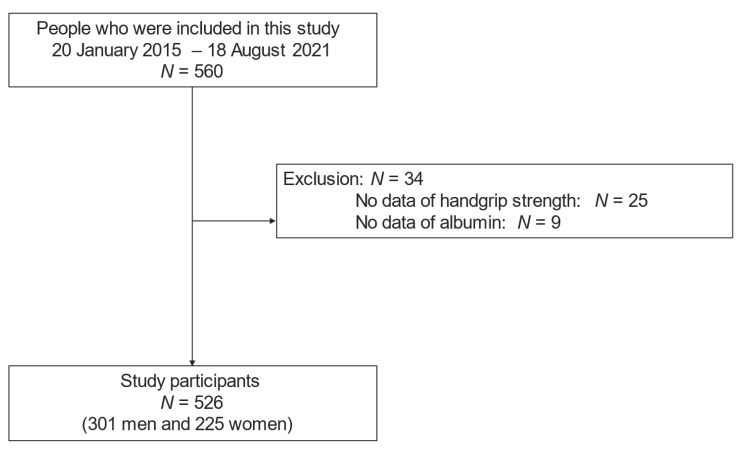
Study flow diagram of the people registration process.

**Figure 2 nutrients-13-03729-f002:**
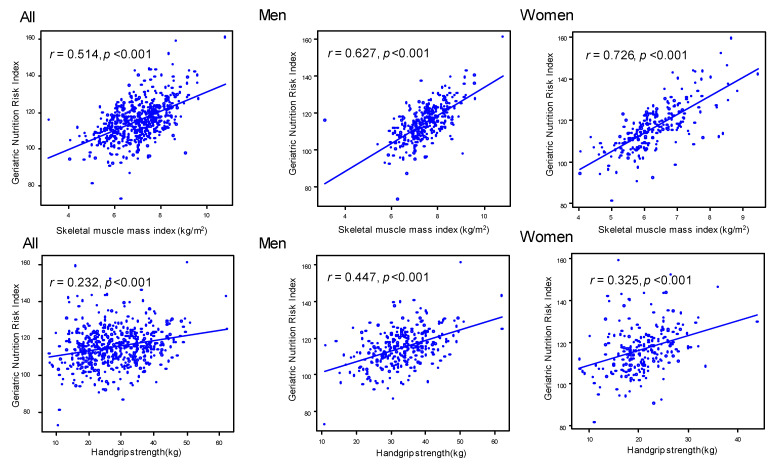
Correlation between geriatric nutrition risk index and skeletal muscle mass index or handgrip strength.

**Table 1 nutrients-13-03729-t001:** Clinical characteristics of study participants.

	All *N* = 526	Men *N* = 301	Women *N* = 225	*p*
Age (years)	67.1 (10.9)	67.7 (10.9)	66.2 (10.8)	0.105
Duration of diabetes (years)	14.6 (10.2)	15.2 (9.8)	13.6 (10.6)	0.076
Family history of diabetes (−/+)	302/224	186/115	116/109	0.024
Height (cm)	161.1 (8.9)	166.9 (5.9)	153.4 (5.8)	<0.001
Body weight (kg)	63.4 (12.6)	66.9 (11.6)	58.8 (12.5)	<0.001
Body mass index (kg/m^2^)	24.4 (4.3)	23.9 (3.6)	25.0 (5.0)	0.007
SBP (mmHg)	133.1 (18.2)	132.1 (17.3)	134.5 (19.2)	0.132
DBP (mmHg)	76.5 (12.4)	76.6 (12.3)	76.4 (12.4)	0.822
Insulin (−/+)	397/129	228/73	169/56	0.948
Multi-injection treatment with insulin (−/+)	499/27	287/14	212/13	0.704
SGLT2 inhibitor (−/+)	431/95	247/54	184/41	1.000
GLP-1 antagonist (−/+)	443/83	260/41	183/42	0.147
Antihypertensive drugs (−/+)	241/285	132/169	109/116	0.339
Presence of hypertension (−/+)	178/348	103/198	75/150	0.905
Smoking (−/+)	449/77	239/62	210/15	<0.001
Habit of exercise (−/+)	276/250	150/151	126/99	0.189
HbA1c (mmol/mol)	57.1 (13.3)	57.1 (12.8)	57.1 (13.9)	0.976
HbA1c (%)	7.4 (1.2)	7.4 (1.2)	7.4 (1.3)	0.976
Plasma glucose (mmol/L)	8.3 (2.7)	8.5 (2.7)	8.0 (2.6)	0.065
Creatinine (umol/L)	74.0 (32.5)	85.2 (36.0)	59.1 (18.7)	<0.001
eGFR (mL/min/1.73 m^2^)	69.7 (19.7)	67.6 (20.2)	72.5 (18.7)	0.005
Uric acid (umol/L)	309.9 (75.7)	327.3 (77.1)	286.7 (67.4)	<0.001
Triglycerides (mmol/L)	1.5 (0.9)	1.5 (1.0)	1.4 (0.8)	0.165
HDL cholesterol (mmol/L)	1.5 (0.4)	1.5 (0.4)	1.6 (0.4)	<0.001
C-reactive protein (ug/L)	2869.9 (10,853.1)	2544.0 (9336.2)	3295.4 (12571.5)	0.444
Albumin (mg/L)	42.5 (3.5)	42.7 (3.6)	42.4 (3.3)	0.330
GNRI	115.5 (10.5)	114.8 (9.9)	116.5 (11.2)	0.058
GNRI score (98</≤98)	27/499	19/282	8/217	0.223
Body fat mass (kg)	19.1 (8.4)	17.5 (7.5)	21.2 (9.1)	<0.001
Percent body fat mass (%)	29.4 (8.8)	25.4 (7.2)	34.7 (7.9)	<0.001
Appendicular muscle mass (kg)	18.3 (4.1)	20.9 (3.1)	14.9 (2.6)	<0.001
SMI (kg/m^2^)	7.0 (1.0)	7.5 (0.8)	6.3 (0.9)	<0.001
Low skeletal muscle mass (−/+)	398/128	223/78	175/50	0.382
Handgrip strength (kg)	27.8 (9.1)	33.1 (7.7)	20.7 (5.2)	<0.001
Low muscle strength (−/+)	395/131	231/70	164/61	0.363
Presence of sarcopenia (−/+)	459/67	259/42	200/25	0.404
	**All** ***N* = 452**	**Men** ***N* = 256**	**Women** ***N* = 196**	** *p* **
Total energy intake (kcal/day)	1747.1 (626.9)	1921.2 (630.4)	1519.7 (545.4)	<0.001
Energy intake (kcal/IBW/day)	30.6 (10.7)	31.4 (10.6)	29.5 (10.8)	0.060
Total protein intake (g/day)	74.0 (31.4)	78.4 (32.3)	68.3 (29.3)	<0.001
Protein intake (g/IBW/day)	1.3 (0.6)	1.3 (0.6)	1.3 (0.6)	0.425
Total fat intake (g/day)	56.3 (23.2)	60.1 (22.9)	51.3 (22.8)	<0.001
Fat intake (g/IBW/day)	1.0 (0.4)	1.0 (0.4)	1.0 (0.5)	0.722
Total carbohydrate intake (g/day)	218.1 (83.6)	238.8 (86.8)	191.1 (70.7)	<0.001
Carbohydrate intake (g/IBW/day)	3.8 (1.4)	3.9 (1.5)	3.7 (1.4)	0.132

Data were expressed as mean (standard deviation) or number. SBP, systolic blood pressure; DBP, diastolic blood pressure; eGFR, estimated glomerular filtration rate; HDL, high-density lipoprotein; GNRI, geriatric nutritional risk index; SMI, skeletal muscle mass index; IBW, ideal body weight.

**Table 2 nutrients-13-03729-t002:** Clinical characteristics according to geriatric nutritional risk index.

	GNRI ≥ 98 *N* = 499	GNRI < 98 *N* = 27	*p*
Age (years)	66.7 (10.9)	73.2 (7.9)	0.002
Sex (men/women)	282/217	19/8	0.223
Duration of diabetes (years)	14.5 (10.1)	16.1 (11.5)	0.417
Family history of diabetes (−/+)	286/213	16/11	1.000
Height (cm)	161.1 (9.0)	160.9 (5.9)	0.872
Body weight (kg)	64.1 (12.5)	50.1 (7.3)	<0.001
Body mass index (kg/m^2^)	24.7 (4.2)	19.3 (2.5)	<0.001
SBP (mmHg)	133.1 (17.8)	133.7 (24.3)	0.877
DBP (mmHg)	76.8 (12.0)	70.9 (17.4)	0.015
Insulin (−/+)	382/117	15/12	0.025
Multi-injection treatment with insulin (−/+)	478/21	21/6	<0.001
SGLT2 inhibitor (−/+)	405/94	26/1	0.083
GLP-1 antagonist (−/+)	419/80	24/3	0.680
Antihypertensive drugs (−/+)	229/270	12/15	1.000
Presence of hypertension (−/+)	170/329	8/19	0.790
Smoking (−/+)	428/71	21/6	0.387
Habit of exercise (−/+)	257/242	19/8	0.087
HbA1c (mmol/mol)	56.6 (12.2)	66.5 (25.0)	<0.001
HbA1c (%)	7.3 (1.1)	8.2 (2.3)	<0.001
Plasma glucose (mmol/L)	8.2 (2.5)	9.9 (4.6)	0.001
Creatinine (umol/L)	73.1 (30.2)	90.6 (59.4)	0.006
eGFR (mL/min/1.73 m^2^)	70.0 (19.4)	64.3 (24.9)	0.145
Uric acid (mmol/L)	311.3 (75.1)	283.5 (82.5)	0.063
Triglycerides (mmol/L)	1.5 (0.9)	1.2 (0.6)	0.127
HDL cholesterol (mmol/L)	1.5 (0.4)	1.6 (0.6)	0.420
C-reactive protein (ug/L)	2115.7 (7863.2)	16,025.9 (30,722.4)	<0.001
Albumin (mg/L)	42.9 (3.0)	35.4 (4.3)	<0.001
GNRI score	116.7 (9.4)	94.1 (5.7)	<0.001
Body fat mass (kg)	19.6 (8.3)	10.1 (5.0)	<0.001
Percent body fat mass (%)	29.9 (8.5)	19.6 (8.3)	<0.001
Appendicular muscle mass (kg)	18.4 (4.2)	16.4 (3.4)	0.016
SMI (kg/m^2^)	7.0 (1.0)	6.3 (1.0)	<0.001
Low skeletal muscle mass (−/+)	389/110	9/18	<0.001
Handgrip strength (kg)	28.0 (9.1)	23.9 (7.9)	0.022
Low muscle strength (−/+)	384/115	11/16	<0.001
Presence of sarcopenia (−/+)	444/55	15/12	<0.001
	**GNRI ≥ 98** ***N* = 433**	**GNRI < 98** ***N* = 19**	** *p* **
Total energy intake (kcal/day)	1740.9 (625.5)	1888.7 (659.6)	0.315
Energy intake (kcal/IBW/day)	30.5 (10.7)	33.0 (10.6)	0.324
Total protein intake (g/day)	74.0 (31.6)	74.8 (27.5)	0.909
Protein intake (g/IBW/day)	1.3 (0.6)	1.3 (0.5)	0.940
Total fat intake (g/day)	56.4 (23.5)	53.3 (15.4)	0.573
Fat intake (g/IBW/day)	1.0 (0.4)	0.9 (0.3)	0.577
Total carbohydrate intake (g/day)	216.8 (82.9)	247.7 (94.7)	0.115
Carbohydrate intake (g/IBW/day)	3.8 (1.4)	4.3 (1.5)	0.125

Data were expressed as mean (standard deviation) or number. The difference between group was evaluated by Student’s *t*-test or chi-square test. SBP, systolic blood pressure; DBP, diastolic blood pressure; eGFR, estimated glomerular filtration rate; HDL, high-density lipoprotein; GNRI, geriatric nutritional risk index; SMI, skeletal muscle mass index; IBW, ideal body weight.

**Table 3 nutrients-13-03729-t003:** Odds ratio of low geriatric nutritional risk index on the presence of sarcopenia.

	Model 1		Model 2		Model 3		Model 4	
	OR (95% CI)	*p*	OR (95% CI)	*p*	OR (95% CI)	*p*	OR (95% CI)	*p*
Low GNRI (<98)	6.46 (2.88–14.5)	<0.001	4.56 (1.90–11.0)	<0.001	5.44 (2.17–13.6)	<0.001	4.88 (1.88–12.7)	0.001
Age (years)	-	-	1.13 (1.08–1.17)	<0.001	1.11 (1.07–1.16)	<0.001	1.11 (1.06–1.16)	<0.001
Sex (women)	-	-	0.99 (0.56–1.75)	0.971	0.99 (0.55–1.78)	0.966	0.95 (0.52–1.73)	0.865
Duration of diabetes (years)	-	-	-	-	1.04 (1.02–1.07)	<0.001	1.05 (1.02–1.07)	0.001
Habit of exercise	-	-	-	-	1.30 (0.73–2.30)	0.374	1.37 (0.77–2.46)	0.288
Habit of smoking	-	-	-	-	0.70 (0.28–1.77)	0.449	0.77 (0.30–1.98)	0.589
HbA1c (mmol/mol)	-	-	-	-	-	-	1.01 (0.99–1.04)	0.345
Insulin treatment	-	-	-	-	-	-	0.82 (0.41–1.65)	0.581
Usage of SGLT2 inhibitor	-	-	-	-	-	-	0.58 (0.22–1.56)	0.282
Usage of GLP-1 antagonist	-	-	-	-	-	-	0.76 (0.28–2.09)	0.593

Model 1 is unadjusted; Model 2 is adjusted for age, sex; Model 3 is adjusted for age, sex, duration of diabetes, habit of exercise, habit of smoking; Model 4 is adjusted for age, sex, duration of diabetes, habit of exercise, habit of smoking, HbA1c, insulin treatment, usage of SGLT2 inhibitor, usage of GLP-1 antagonist. GNRI, geriatric nutritional risk index.

**Table 4 nutrients-13-03729-t004:** Odds ratio of the continuous geriatric nutritional risk index on the presence of sarcopenia.

	Model 1		Model 2		Model 3		Model 4	
	OR (95% CI)	*p*	OR (95% CI)	*p*	OR (95% CI)	*p*	OR (95% CI)	*p*
GNRI	0.88 (0.85–0.91)	<0.001	0.89 (0.86–0.93)	<0.001	0.89 (0.86–0.93)	<0.001	0.89 (0.86–0.93)	<0.001
Age (years)	-	-	1.10 (1.06–1.15)	<0.001	1.09 (1.04–1.14)	<0.001	1.09 (1.04–1.14)	<0.001
Sex (women)	-	-	0.98 (0.54–1.77)	0.930	0.92 (0.50–1.70)	0.792	0.88 (0.47–1.64)	0.690
Duration of diabetes (years)	-	-	-	-	1.04 (1.01–1.06)	0.007	1.04 (1.01–1.07)	0.006
Habit of exercise	-	-	-	-	1.13 (0.62–2.03)	0.695	1.17 (0.64–2.13)	0.618
Habit of smoking	-	-	-	-	0.58 (0.22–1.56)	0.280	0.59 (0.21–1.65)	0.315
HbA1c (mmol/mol)	-	-	-	-	-	-	1.01 (0.98–1.03)	0.663
Insulin treatment	-	-	-	-	-	-	0.70 (0.34–1.43)	0.325
Usage of SGLT2 inhibitor	-	-	-	-	-	-	0.83 (0.29–2.37)	0.732
Usage of GLP-1 antagonist	-	-	-	-	-	-	0.99 (0.34–2.87)	0.991

Model 1 is unadjusted; Model 2 is adjusted for age, sex; Model 3 is adjusted for age, sex, duration of diabetes, habit of exercise, habit of smoking; Model 4 is adjusted for age, sex, duration of diabetes, habit of exercise, habit of smoking, HbA1c, insulin treatment, usage of SGLT2 inhibitor, usage of GLP-1 antagonist. GNRI, geriatric nutritional risk index.

## Data Availability

The datasets generated during and/or analyzed during the current study are available from the corresponding author on reasonable request.

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
