# Peer review of "Association between Geriatric Nutrition Risk Index and The Presence of Sarcopenia in People with Type 2 Diabetes Mellitus: A Cross-Sectional Study"

_nutrients, 2021, doi:10.3390/nu13113729_

Round 1
Reviewer 1 Report
The article needs a minor revision of English.
The work and statistical analysis are well explained.
There are any data regarding the specific diet followed by the patients?
From the data described, patients with lower GNRI had a higher HbA1c.
Were there patients exclusively on multi-injection treatment with insulin?
If so, how many were those with GNRI <98?
Are there any data regarding the concentration of the C-peptide between the two groups? If yes
there was a difference between the two groups?
Author Response
Point 1
The article needs a minor revision of English.
Response
Thank you for your comments. According to your comments, we have revised this article.
Point 2
The work and statistical analysis are well explained.
Response
Thank you for your comments.
Point 3
There are any data regarding the specific diet followed by the patients?
Response
Thank you for your comment. As your say, it has been reported that nutritional status and habitual dietary intake was closely related. According to your comment, we have analyzed the association between habitual diet intake and GNRI among the participants with habitual diet intake data (n = 452/ 562). Habitual dietary intakes were not associated with the GNRI. This might be because that this study was the cross-sectional study and that the participants were already received nutritional guidance. We have mentioned these points in the Methods, Results and Discussion sections.
Methods (Line: 96-100)
“Dietary intake was evaluated by brief-type self-administered diet history questionnaire (BDHQ), which estimated a dietary intake of 58 items over the past month [14]. Total energy (kcal/day); total protein (g/day); fat (g/day); and carbohydrate (g/day) intakes were obtained by the BDHQ. Total energy (kcal/IBW/day), total protein (g/IBW/day), fat (g/IBW/day), and carbohydrate (g/IBW/day) intakes were calculated.”
Results (Line: 138-140)
“In addition, among 526 participants, 452 participants (433 participants with high GNRI and 19 participants with low GNRI) were surveyed about their dietary intake.”
Results (Line: 166-167)
“There was no difference of habitual diet intakes between participants with high GHRI and low GNRI.”
Discussion (Line: 233-236)
“Habitual dietary intakes were not associated with the GNRI, although it has been reported that nutritional status and habitual dietary intake was closely related [28]. This might be because that this study was the cross-sectional study and that the participants were already received nutritional guidance.”
Reference
- Kobayashi, S.; Honda, S.; Murakami, K.; Sasaki, S.; Okubo, H.; Hirota, N.; Notsu, A.; Fukui, M.; Date, C. Both Comprehensive and Brief Self-Administered Diet History Questionnaires Satisfactorily Rank Nutrient Intakes in Japanese Adults. Journal of Epidemiology 2012, 22, 151–159, doi:10.2188/jea.JE20110075.
- 28. Katano, S.; Hashimoto, A.; Ohori, K.; Watanabe, A.; Honma, R.; Yanase, R.; Ishigo, T.; Fujito, T.; Ohnishi, H.; Tsuchihashi, K.; et al. Nutritional Status and Energy Intake as Predictors of Functional Status after Cardiac Rehabilitation in Elderly Inpatients with Heart Failure ― A Retrospective Cohort Study. Circulation Journal 2018, 82, 1584–1591, doi:10.1253/circj.CJ-17-1202.
Point 4
From the data described, patients with lower GNRI had a higher HbA1c.
Were there patients exclusively on multi-injection treatment with insulin?
If so, how many were those with GNRI <98?
Response
Thank you for your valuable comment. In this study, the proportion of patients exclusively on multi-injection treatment with insulin was 5.1% (n = 27/526). The proportion of patients exclusively on multi-injection treatment with insulin in low GNRI was higher than that in high GNRI (22.2% [n = 6/27] vs. 4.2% [n = 21/499], p <0.001). Therefore, we have described this point in the Results section as below.
Results (Line: 161-163)
“The proportion of patients exclusively on multi-injection treatment with insulin in low GNRI was higher than that in high GNRI (22.2% [n = 6/27] vs. 4.2% [n = 21/499], p <0.001).”
Point 5
Are there any data regarding the concentration of the C-peptide between the two groups? If yes there was a difference between the two groups?
Response
Thank you for your comment. As you say, the difference of the concentration of C-peptide between the two groups might be an important issue. Unfortunately, however, we did not measure the concentrations of C-peptide. Therefore, we have mentioned this point as one of the limitations of this study in the Discussion section described as below.
Discussion (Line: 240-241)
“Fourth, we did not assess the concentration of C-peptide in this study. Therefore, this study did not consider the capacity for insulin secretion.”
Reviewer 2 Report
Please consider making the following changes to address grammar and spelling issues.
Line 24: Please change to “WAS diagnosed as sarcopenia” (Having both … was diagnosed)
Line 31: Would clarify whether these correlations were positive or negative
Line 48: Change to “than THOSE without diabetes”
Lines 72-73: Change to “people with T2DM whose body composition was measured” or “ “people with T2DM with body composition measurements”
Line 93: Remove “or” from “defined as or usage”
Line 101: Change to “divided BY height squared”
Line 107: Change to “WAS classified as”
Lines 119-120: Change to “Differences in continuous and categorical variables”
Figure 1: Albumin is misspelled; men is also misspelled
Line 135: Change “people” to “participant”
Line 139: Change to “mean BMI WAS”
Line 145: “Data WERE expressed” (please change all instances of this, as data is a plural word)
Line 171: “Data WERE expressed”
Line 184: removed “-” between low-Geriatric
Lines 217-218: Change to “malnutrition is a major risk factor for sarcopenia"
Author Response
Point 1
Please consider making the following changes to address grammar and spelling issues.
Response
Thank you for your suggestions. According to your suggestion, we have revised them.
Point 2
Line 24: Please change to “WAS diagnosed as sarcopenia” (Having both … was diagnosed)
Response
Thank you for your comment. According to your comment, we have revised it in the Abstract section as below.
Abstract (Line: 23-24)
“Having both low handgrip strength (<28 kg for men and <18 kg for women) and low skeletal muscle mass index (<7.0 kg/m2 for men and <5.7 kg/m2 for women) was diagnosed as sarcopenia.”
Point 3
Line 31: Would clarify whether these correlations were positive or negative
Response
Thank you for your comment. Both handgrip strength (r = 0.232, p <0.001) and skeletal muscle mass index (r = 0.514, p <0.001) were positively correlated with the GNRI. Therefore, we have revised it in the Abstract section as below.
Abstract (Line: 30-32)
“The GNRI showed positive correlations with handgrip strength (r = 0.232, p <0.001) and skeletal muscle mass index (r = 0.514, p <0.001).”
Point 4
Line 48: Change to “than THOSE without diabetes”
Response
Thank you for your comment. According to your comment, we have revised it in the Introduction section as below.
Introduction (Line: 47-48)
“Therefore, sarcopenia in people with T2DM requires more attention than those without diabetes.”
Point 5
Lines 72-73: Change to “people with T2DM whose body composition was measured” or “ “people with T2DM with body composition measurements”
Response
Thank you for your comment. According to your comment, we have revised it in the Methods section as below.
Methods (Line: 71-72)
“In this study, people with T2DM with body composition measurements from January 2015 to August 2021 were included.”
Point 6
Line 93: Remove “or” from “defined as or usage”
Response
Thank you for your comment. According to your comment, we have revised it in the Methods section as below.
Methods (Line: 93-95)
“Hypertension was defined as usage of antihypertensive drugs, systolic blood pressure of ≥140 mmHg, and/or diastolic blood pressure of ≥90 mmHg [13].”
Point 7
Line 101: Change to “divided BY height squared”
Response
Thank you for your comment. According to your comment, we have revised it in the Methods section as below.
Methods (Line: 105-107)
“Then, body mass index (BMI, kg/m2) or skeletal muscle mass index (SMI, kg/m2) was estimated by BW (kg) or appendicular muscle mass (kg) divided by height squared (m2) [15].”
Point 8
Line 107: Change to “WAS classified as”
Response
Thank you for your comment. According to your comment, we have revised it in the Methods section as below.
Methods (Line: 111-114)
“Having both low muscle strength (handgrip strength: <28 kg for men and <18 kg for women) and low skeletal muscle mass (SMI: <7.0 kg/m2 for men and <5.7 kg/m2 for women) was classified as sarcopenia [2].”
Point 9
Lines 119-120: Change to “Differences in continuous and categorical variables”
Response
Thank you for your comment. According to your comment, we have revised it in the Methods section as below.
Methods (Line: 124-125)
“Differences in continuous and categorical variables were evaluated using the Student’s t-test and chi-squared test, respectively.”
Point 10
Figure 1: Albumin is misspelled; men is also misspelled
Response
Thank you for your comment. According to your comment, we have revised them.
Point 11
Line 135: Change “people” to “participant”
Response
Thank you for your comment. According to your comment, we have revised it in the Results section as below.
Results (Line: 136-138)
“We excluded 34 participants, of whom 25 were not assessed for handgrip strength, while 9 were not assessed for serum albumin levels; therefore, the final study population comprised 526 participants (301 men and 225 women; Figure 1).”
Point 12
Line 139: Change to “mean BMI WAS”
Response
Thank you for your comment. According to your comment, we have revised it in the Results section as below.
Results (Line: 144-145)
“The mean age or BMI was 67.1±10.9 years or 24.4±4.3 kg/m2 in all participants.”
Point 13
Line 145: “Data WERE expressed” (please change all instances of this, as data is a plural word)
Line 171: “Data WERE expressed”
Response
Thank you for your comments. According to your comments, we have revised them in the Results section as below.
Results (Line: 152 and 181)
“Data were expressed as mean (standard deviation) or number.”
Point 14
Line 184: removed “-” between low-Geriatric
Response
Thank you for your comment. According to your comment, we have revised it as below.
Table 3. (Line: 194)
“Odds ratio of low geriatric nutritional risk index on the presence of sarcopenia.”
Point 15
Lines 217-218: Change to “malnutrition is a major risk factor for sarcopenia"
Response
Thank you for your comment. According to your comment, we have revised it in the Discussion section as below.
Discussion (Line: 228-230)
“Moreover, malnutrition is a major risk factor for sarcopenia and plays as a driver of loss of muscle mass and function, which are the main features [5,26].”